# The Application and Challenge of Binder Jet 3D Printing Technology in Pharmaceutical Manufacturing

**DOI:** 10.3390/pharmaceutics14122589

**Published:** 2022-11-24

**Authors:** Xuejun Chen, Shanshan Wang, Jie Wu, Shuwei Duan, Xiaolong Wang, Xiaoxuan Hong, Xiaolu Han, Conghui Li, Dongzhou Kang, Zengming Wang, Aiping Zheng

**Affiliations:** 1Pharmaceutical Experiment Center, College of Pharmacy, Yanbian University, Yanji 133002, China; 2State Key Laboratory of Toxicology and Medical Countermeasures, Beijing Institute of Pharmacology and Toxicology, Beijing 100850, China; 3College of Biotechnology, Tianjin University of Science and Technology, Tianjin 300457, China; 4Department of Nephrology, First Medical Center of Chinese PLA General Hospital, National Clinical Research Center for Kidney Diseases, Beijing 100853, China

**Keywords:** binder jet 3D printing, computer-aided design, pharmaceutical manufacturing, drug delivery system, mechanism

## Abstract

Three-dimensional (3D) printing is an additive manufacturing technique that creates objects under computer control. Owing to the rapid advancement of science and technology, 3D printing technology has been widely utilized in processing and manufacturing but rarely used in the pharmaceutical field. The first commercial form of Spritam^®^ immediate-release tablet was approved by FDA in 2015, which promoted the advancement of 3D printing technology in pharmaceutical development. Three-dimensional printing technology is able to meet individual treatment demands with customized size, shape, and release rate, which overcomes the difficulties of traditional pharmaceutical technology. This paper intends to discuss the critical process parameters of binder jet 3D printing technology, list its application in pharmaceutical manufacturing in recent years, summarize the still-open questions, and demonstrate its great potential in the pharmaceutical industry.

## 1. Introduction

Throughout history, civilization has undergone recurrent transformations. Nowadays, a new technology revolution represented by artificial intelligence and 3D printing has propelled the world into a new wave of global innovation boom [1]. Three-dimensional printing was first proposed by engineer Charles Hull in the early 1980s [2]. It is a manufacturing process based on a pre-designed 3D digital model and then through layer by cumulative layer printing of materials until the construction of three-dimensional entities is completed. Therefore, 3D printing is also termed additive manufacturing (AM) or layered manufacturing [3]. With the progress of science and technology, 3D printing technology has shown a diversified development tendency. According to the American Society for testing and materials (ASTM) classification, there are several additive manufacturing technologies and molding processes, including binder jetting, vat photopolymerization, material jetting, material extrusion, powder bed fusion, sheet lamination, and directed energy deposition [4]. Due to 3D printing technology’s enormous advantages in producing customized products with complex shapes and structures [5] has rapidly developed in automotive, electronics, aviation, cell engineering, bio-medical engineering, etc. [6].

Additionally, to analyze the relationship between 3D printing, pharmacy, and orthopedics, we used VOS viewer to construct a network visualization map (Figure 1). The co-word clustering analysis is performed by the VOS viewer, and the “Label View” is used to display the co-word clustering. In the Label View, each node represents a keyword or term. The higher the frequency of its occurrence, the larger the node will be. One color represents a cluster, and a cluster is composed of keywords or terms with high co-occurrence. The connecting line in the figure means that two keywords connected have at least one co-occurrence in the literature. The higher the frequency of keyword co-occurrence is, the thicker the line will be. According to the keyword analysis map, it is evident that binder jet 3D printing is a hot spot topic in the field of 3D printing in pharmacy and orthopedics.

In 2015, the FDA approved the world’s first 3D printing formulation, indicating that binder jet 3D printing has been integrated into pharmaceutical manufacturing and applied to pharmaceutical manufacturing. Up to now, binder jet 3D printing technology is still the only 3D printing process approved by FDA. Compared with other 3D printing technologies, one important advantage of the binder jet 3D printing technology is its potential to realize full-color printing. In addition, it has less stringent requirements for raw and auxiliary materials and does not need support in the printing process, resulting in efficient and quick printing [7]. Binder jet 3D printing is considered to be the most suitable 3D printing technology for industrialization. Although it is still in its infancy in pharmaceutical manufacturing, there is no doubt that binder jet 3D printing will improve the level of implants and pharmaceutical manufacturing and will have the potential to enable multi-tablet formulation, release kinetic control, and realize personalized administration [8], leading to the development of customized drug delivery systems. This review introduces the binder jet 3D printing technology and expounds on the factors affecting binder jet 3D printing technology. This paper lists the relevant applications of binder jet 3D printing in pharmaceutical manufacturing to provide new ideas for binder jet 3D printing technology in pharmaceutical design and manufacturing. Finally, the prospects and challenges of binder jet 3D printing technology are discussed.

## 2. Binder Jet 3D Printing

Binder jet 3D printing was invented at the Massachusetts Institute of Technology (MIT) and patented by Emanuel Sachs in 1993. It is a powder-based 3D printing process. The binder is sprayed onto the powder in tiny ink droplets via the printing nozzle, bonding the powder to form a 3D printing structure [9]. As a result, the complete binder jet 3D printing system is composed of the following components: a model design software, a printing nozzle, a powder spreading roller, an ink cartridge containing printing liquid, a powder supply platform, a printing platform, and a powder collection device. The process for printing formulation is as follows: 3D object design is created by computer-aided design (CAD) software which is further converted to .stl or some other format compatible with printers. Before printing .stl file is subjected to slicing, which converts the object to 2D layers and creates instructions for printing in G-code.

Moreover, the powder is evenly spread on the printer console using a computer-controlled powder spreading roller. The printing head sprays binder or medication-containing ink droplets onto the powder bed at a specific rate along a predetermined path. The console then descends, the powder supply platform rises, the powder spreading roller rolls to re-spread the powder, and the printing nozzle sprays droplets [10]. The final preparations are manufactured in this manner, utilizing the “layered manufacturing and layer-by-layer superposition” principle (Figure 2). The powder that was not sprayed during the printing process can be used to create a backing material for printed products. After printing, this powder can be collected for future use [11].

### 2.1. Factors Affecting the Binder Jet 3D Printing Process

The printing process for binder jet 3D printing is divided into three stages: the first stage involves the generation of ink droplets; the second stage consists of depositing ink droplets and interacting with powder; the third stage includes curing and removing powder [13]. When using the binder jet 3D printing process for manufacturing drugs, the factors in connection with the print head, binder, powder, and printing parameters should be understood.

#### 2.1.1. The Print Heads

The inkjet 3D printing technique can be classified into two types based on the creation and positioning of ink droplets: continuous inkjet printing (CIJ) and drop-on-demand (DoD) [14]. In the CIJ approach, ink droplets are continually ejected from the printing nozzle and discharged onto the powder material. CIJ printing is utilized frequently for coding and marking applications. In the DoD method, several ink droplets are ejected from the printing nozzle, which is dominant in graphics and text printing [15]. Binder jet 3D printing is a DoD technique using a thermal bubble print nozzle or a piezoelectric print nozzle (Figure 3). In the thermal bubble DoD, the heating element inserted in the nozzle produces bubbles. The binder is locally heated, and ink droplets are ejected during the printing process due to the pressure differential [16]. The piezoelectric printing nozzle uses the piezoelectric element to receive the electrical signals and generate pressure in the piezoelectric DoD. The ink droplets can overcome surface tension and be ejected from the printing nozzle. The piezoelectric printing nozzle receives an electrical input and deforms to generate pressure, allowing the ink droplets to overcome their surface tension and be ejected from the printing nozzle [9]. The thermal print head is more suitable for an aqueous-based binder. However, nozzles are prone to corrosion and clogging due to prolonged high-temperature and high-pressure working conditions. In comparison to the stringent requirements for thermal bubble printing nozzles due to the volatility of the printing liquid, piezoelectric printing nozzles have a broader applicability range for printing liquid [17]. In addition, the piezoelectric printing nozzle has a more robust control ability on ink droplets. The micro shape of ink droplets is more regular than the thermal print head, and the positioning is more accurate. By controlling the voltage to adjust the size and use mode of ink droplets effectively, higher printing accuracy and printing effect can be obtained. The printing speed, direction, nozzle diameter, and resonance frequency of the printing nozzle all affect its performance, which impacts the formability of the printing preparation [18].

#### 2.1.2. Binder Solution

Binder solution plays a critical role in the printing process. It can fill the gap between powder materials and create the desired shape [9]. The binder system may be aqueous, organic, or a mixture of two or more organic solvents. It is worth noting that the toxicity of the binder should be as low as possible. Ideally, the binder’s components should be soluble and compatible with each other. If the printing nozzle is not clogged, insoluble components with colloidal particle sizes can be added to the binder [20]. It mainly affects the printing molding from three aspects: the generation of droplets in the printing process, the impact, rebound, and diffusion of droplets on the powder bed, and the drying and solidification after printing. The generation of droplets is related to the type of printing nozzle. The binder’s viscosity, density, and surface tension will affect the consistency and stability of the droplet formation process, thus affecting the printing effect [21]. The printability property of the binder solution can be defined as the ability of the binder to produce stable droplets on every occurrence, which is indispensable to the printing process. It can be characterized by several dimensionless groupings of physical constants, the most useful of which are the Reynolds (Re), Weber (We), and Ohnesorge (Oh) numbers [22]:(1)Calculation of Reynolds number to measure the relative magnitude of fluid inertia force and viscous force:
(1)Re=vρaη
(2)Calculation of Weber number representing the ratio of inertial force to surface tension effect:(2)$$We=\frac{v^{2}\rho a}{\gamma }$$
(3)Oh=ηγρa=WeRe
(4)$$Z=\frac{1}{Oh}$$
where *ρ*, *η*, and *γ* are the density, dynamic viscosity, and surface tension of the binder, respectively, v is the velocity, and a is a characteristic length.

Fromm [23], characterized drop formation and utilized another parameter, Z, which is 1/Oh, to select the appropriate binder. He proposed that the fluids of Z > 2 could produce stable drops or were jettable. Reis and Derby [24] further optimized the range of Z values. When 1 < Z < 10, stable droplets could be produced. Experiments by Remi Noguera and colleagues [25] showed that viscous dissipation would hinder droplet injection when the Z value was low (Z < 1). To maintain a high injection speed in the injection process, higher pressure was needed to overcome viscous dissipation. When the Z value was high (such as Z > 10), the main droplet would be accompanied by many satellite droplets, reducing printing resolution and precision (Figure 4). Jang et al. [26] redefined the range of the values for Z for good printability. Inks with Z < 4 already resulted in long-lived filaments during the drop formation. When the Z value exceeded 14, the ink could not form single drops during printing due to the low viscosity. Zhong et al. [27] further limited the range of Z by settling on the Z value between 4 and 8.

The binder system’s surface tension should be high enough to form droplets and prevent binder leakage when the print nozzle is not used. Second, the viscosity of the binder plays a significant role in the preparation’s formation. If the viscosity is too high, there will be a high injection pressure, resulting in the uneven injection of the printing liquid and nozzle blockage. If the viscosity is too low, satellite droplets will form, and preparations will print unevenly [28,29]. The amount of binder is also a factor affecting the printing process. Too much binder will stick to excess powders, resulting in an incorrect preparation size. When the amount of binder is insufficient, the preparation has poor mechanical properties [30].

#### 2.1.3. Powder-Specific Properties

The foundation of the binder jet 3D printing is powder material. Powder-specific properties strongly influence many aspects of the binder jetting process. Therefore, it is vital to ensure the uniform distribution and packing of the powder in the powder bed. The significant characteristics of powder typically studied in binder jet 3D printing include:(1)powder particle characteristics (including surface morphology and powder particle size and size distribution),(2)powder flowability,(3)powder packing density.

Chen et al. [31] mentioned that spherical shape particles were the first choice in binder jet 3D printing. It could reduce the tendency of mechanical interlocking and the force of friction between particles to form a consistent powder flow, which results in an even distribution of the powder on the powder bed. Li et al. [32] explored the effect of force between particles on powder flowability. The experiment confirmed that the van der Waals force between powders with small particle sizes was large. Small particles were more prone to aggregation and poor flowability, negatively impacting powder packing density. Large particles had good flowability but will affect the uniformity of powder lay-downing and printing accuracy in the wettability and adhesion of the powder. On the other hand, the smaller the average particle size was, the higher the printing resolution would be [33]. The influence of particle size and particle size distribution of powder on the printing effect was evident, which could directly interfere with the mechanical strength and surface roughness of 3D printing preparation [34]. The flowability of the powder in binder jet 3D printing has a significant impact on obtaining high-resolution printing. A schematic illustration of the main stages of single binder drop powder interaction during 3DP is shown in Figure 5. Butscher et al. [35] indicated that powder flowability was essential for building thin powder layers (steps 1 and 6) and removing the powder from the printing part (steps 8). When powder flowability is sufficient, it might not be able to prepare a complete preparation. For example, Butscher et al. [36] observed that large calcium phosphate powder showed excellent flow behavior but the layers displaced during the printing process. In comparison, poor flowability would decrease printing resolution [20] and hinder the removal of powder inside the pores. It made de-powdering a problematic and sometimes unachievable goal [35]. In addition, compared with the traditional preparation method, binder jet 3D printing lacks the step of compacting the powder. The powder’s packing density was closely related to the mechanical strength of the preparation. A. Antic et al. [37] pointed out powder wettability’s importance in reflecting printing accuracy. Poor powder wettability prolonged the liquid penetration time, resulting in poor formability of the preparation. Sohn et al. [38] studied the effect of particle size on the powder packing density. Their article showed that the powder packing density was affected by powder size distribution and particle morphology. It was independent of particle size for large particles. Ziegelmeier et al. [39] indicated that when the packing density in the powder bed reached a high level, the number of inter-particle voids in the powder bed was low. Notably, decreasing the porosity will negatively affect binder penetration and, in some cases, may result in inadequate mechanical characteristics of the product.

#### 2.1.4. Printing Parameters

In the binder jet 3D printing process, the printing parameters also play an essential role in the mechanical strength of the printed parts. The printing parameters include the following:(1)the thickness of the powder layer;(2)the pushing speed of the powder roller;(3)the jetting amount or times of the printing liquid;(4)the height of the print head from the powder layer.

Enneti et al. [40] studied the effect of layer thickness and binder dosage on the mechanical strength of the printing product and proved that the binder dosage and powder layer thickness directly affected the strength of the printing product. Increasing the thickness of the powder layer would reduce mechanical strength. While increasing the amount of binder under constant layer thickness would increase the mechanical strength of the preparation. The advancing speed of the powder roller was related to the printing accuracy of the printing product. Chen et al. [31] noticed significant defects on the powder bed that may extend up to a few centimeters at a lower speed. Printing parts did not usually survive during powder spreading. Those defects were aligned perpendicularly to the wiper displacements and created significant gaps in the powder bed, affecting several consecutive layers. It tended to appear immediately after a certain number of layers. When turned to the appropriate roller speed, the lower contact time of the roller with the powder bed resulted in minimized contortion of the powder bed by the roller laterally. They noticed no such defects, and the whole powder bed remained smooth without any crack or defect in the subsequent tests. However, too fast of a pushing speed of the powder roller would lead to uneven distribution of printing powder, which was likely to cause tablet delamination. Therefore, the setting of parameters needs to be adjusted adaptively according to the binder and printing powder [41].

### 2.2. Simulation Study on Binder Jet 3D Printing Process

To date, most of the binder jet 3D printing process parameters have been optimized in a large number of empirical studies, and material and financial resources are often required. Simulations can provide new insights into the complex physics of binder jet 3D printing, which can help guide process modifications and improvements [42].

Wang et al. [43] adopted the numerical simulation method to simulate the formation and fall of binder droplets from the nozzle. The whole droplet generation process was successfully simulated in one whole cycle, including the primary stage, necking stage, break-up stage, the formation stage of satellite drops, and the stage of the main and satellite droplets into one. By analyzing the droplet formation process, the changing pattern of droplet shape can be found, and the quality of binder jet printing can be predicted to some extent.

Powder spreading is critical in binder jet 3D printing. Lee et al. [44] based on the discrete element method (DEM) to simulate the powder spreading process and capture local variations in the powder bed. The influence of roller velocity and RPM on powder bed density, surface roughness, and particle segregation of powder spreading was simulated, respectively. The simulation results showed that roller velocity greatly influenced the formation of homogeneous PSD over the powder bed. In contrast, the powder bed surface roughness was increased. It occurred on unfilled edges and unleveled surfaces during the spreading. A.L. Maximenko et al. [45] worked on the analysis of powder spreading during binder jet 3D printing by the DEM (Figure 6). Both modeling and experimental observations confirmed that the powder spreads in binder jet 3D printing. The displacement occurred in the powder’s upper layers but also in the powder particles that had been bonded in the bottom layer in the previous step. Zhang et al. [46] used DEM to simulate powder spreading by roller. Their experimental results demonstrated that increasing the translational velocity of the roller led to a reduced powder-bed density (Figure 7). The layer thickness was also an important factor affecting the powder-bed density. When the layer thickness was only 50 µm, most particles were pushed out of the building platform, thus forming cavities. The powder bed became more uniform and compact when the layer thickness was more than 150 µm (Figure 8).

In order to improve the forming precision and efficiency of the 3DP process, Deng et al. [47] used the volume-of-fluid (VOF) method to simulate the impact and infiltration processes on the powder bed with the single micrometer droplet. The simulated results showed that the binder droplet had a longer penetration time, a wider spreading diameter, and a smaller penetration depth on the smaller particle size (Figure 9). The comparison with the experimental results proved the feasibility of the simulation in the binder impact process and infiltration process on the powder bed.

## 3. Applications of Binder Jet 3D Printing in Pharmaceutical Manufacturing

Binder jet 3D printing technology has enabled the production of some preparations that are difficult to make using conventional technology. Modifying the model’s structure and shape and adjusting the prescription ratio make it possible to control the drug release mode flexibly. Multiple drugs can be used to treat diseases by separating the preparation space. Moreover, drug delivery can be accomplished by establishing a separate chamber structure. A variety of powdered materials can be used for printing if a suitable binder is used when printing formulations using binder jet 3D printing technology. The Active Pharmaceutical Ingredient (API) is added to the formulation in two ways. One is that the cartridge contains only the solvent and binder, and the API is mixed well with the excipients as a printed powder. The other is that API is sprayed as a solution or nanoparticle suspension onto the powder bed [48].

### 3.1. Orally Rapidly Releasing Dosage Forms

“Immediate release preparations” refers to a solid formulation that disintegrates rapidly after administration. Its advantages include ease of administration, rapid drug absorption, and high bio-availability, which are particularly well-suited for certain medications that require rapid onset of action [49]. Binder jet 3D printing technology is a non-compressible technology created using the “layered manufacturing and layer-by-layer superposition” principle. The obtained preparation typically has a high porosity, facilitating liquid penetration, thereby promoting drug disintegration and accelerating the rate of API release [50].

Cui et al. [51] fabricated high-dose theophylline dispersible tablets and low-dose metoprolol tartrate dispersible tablets by adjusting the model volume and drug loading mode. The dosing accuracy of the printed tablets with target drug content was 91.2~108%, with a slight variation coefficient in the 0.5~3.2%. Each tablet shows excellent mechanical properties and structural integrity. Tablets have a similar release profile to commercial preparations and have little effect on the drug in vitro release behavior. Additionally, this example demonstrates that binder jet 3D printing is more accurate at dose regulation than the conventional dose division method.

Sen et al. [52] used amitriptyline hydrochloride as a model drug, dissolving it in various mass concentrations of deionized distilled water and preparing four binders with varying concentrations. The binder jet 3D printing technology was used to prepare the dispersible tablets, which realized the same tablet model and obtained different drug loading results. The tablets disintegrated completely within 1.5 min in the in vitro dissolution test, and more than 85% of the drugs were released within 30 min.

Compared to conventionally fabricated tablets, high-dose API typically results in technical difficulties during manufacturing and quality control, for it is not easy to achieve a high drug content [53]. The preparation of pharmaceutical preparations with a high drug content is a research area for 3D printing preparations. By increasing the drug loading of preparations, patients can take fewer preparations, which improves compliance [54]. Sophia et al. [55] chose caffeine as the printing powder because the model drug and excipients were thoroughly mixed. Moreover, a 70% (*v/v*) ethanol solution was used as the printing solution. Binder jet 3D printing was used to prepare with a 70% drug load.

Given that nearly 50% of the drugs listed and approximately 70% of the raw materials used in the research are water-insoluble compounds [56], Marta et al. [57] used the hydrophobic API clotrimazole as a model drug and then CO treatment with PVP in a suitable proportion followed by spray drying, thereby increasing the wettability and printability of clotrimazole. The final 3D printing clotrimazole dispersible tablets had uniform content of the drug, excellent mechanical properties, and a highly porous structure resulting in a short disintegration time and fast dissolution rate. This example demonstrates a method for achieving rapid dispersion and dissolution of insoluble drugs. Table 1 summarizes some of the applications of binder jet 3D printing reported in the literature, showing its active performance in pharmaceutical manufacturing.

### 3.2. Sustained-Release Preparations and Controlled-Release Preparations

Slow and controlled release formulations offer the benefits of stable drug release, prolonged efficacy, and few adverse effects. It not only keeps drug concentrations in the blood flow and avoids side effects, it significantly extends the duration of action, reduces the frequency of medication administration, improves patient compliance, and plays an essential role in the production and use of drugs [64].

Wu et al. [65] developed the first binder jet 3D printing controlled-release tablet in 1996, using methylene blue and Alizarin Yellow as model drugs. It is considered unsuitable due to its high toxicity and difficulty in removing traces of the binder from the tablets. Pharmacokinetic theory indicates that the drug release rate is proportional to the geometry of the solid preparation, and changing the geometry of the preparation will affect the drug’s release curve [66]. Yu et al. [67] developed pie-shaped controlled-release tablets with acetaminophen as the model drug. The tablet was divided into three sections. There was no drug on the top and bottom of the pill, preventing water infiltration and drug diffusion. Converting the drug from a typical three-dimensional release mode to a two-dimensional release mode in which the drug was released only from the inner and outer cylindrical surfaces of the tablet. The low-viscosity water-soluble polymer was used as the skeleton material inside the drug-containing cylinder to ensure that the tablets released the drug according to the dissolution mechanism. At the same time, the annular area surrounding the pill contained a high concentration of drug-release inhibiting material. As the drug-release area of the outer cylindrical surface decreases gradually, the drug-release area of the inner hole increases synchronously. Since the tablet’s height remained relatively constant over time, the drug release rate also remained relatively stable. Moreover, Yu et al. [68] used binder jet 3D printing to create an API gradient oral controlled-release drug delivery system and dissolved diclofenac sodium in the binder. The top and bottom layers were intended for release inhibition in the axial direction, while the middle layer powder was a mixture of lactose, HPMC, and PVP. The model was divided laterally into four regions from inside to outside. The drug gradient was changed by decreasing the number of binder sprays in each area from inside to outside. The drug delivery system employed vertical up and down release inhibition to convert the three-dimensional natural release into a radial two-dimensional controlled release. Increasing the radial drug concentration to compensate for the reduced drug release area could achieve the same amount of drug released per unit of time, thereby performing the controlled release function.

### 3.3. Fabrication of Dosage Forms with Multiple Drugs

3D printing enables the production of customized drug dosage forms on demand. It can produce more complex dosage forms, such as oral dosage forms containing a range of APIs. The goal is to increase patient compliance by lowering the daily number of taking medicine [69]. Levetiracetam (LEV), as an antiepileptic drug, effectively treats childhood epilepsy. Children frequently experience neurological and behavioral side effects, which are higher than adults during long-term regular medication. Pyridoxine hydrochloride (PN) has been shown in studies to reduce behavioral side effects associated with LEV use, significantly improve the antiepileptic development of LEV, decrease adverse reactions, and improve the impact on nerves and behavior [70,71]. From the inside to the outside. Hong et al. [72] created a three-layer nested tablet model with a chamber structure, followed by a hollow powder layer, a PN-containing nested layer, and a PN-free shell layer (Figure 10). As printing powder, LEV and other excipients are evenly mixed. The printing solution was divided into blank printing and printing solutions containing PN. A PN printing solution was sprayed into the tablet using binder jet 3D printing technology to prevent drug degradation caused by light and other factors. This experiment involved the preparation of an LEV-PN compound with excellent mechanical properties. The volume of a single-layer inkjet was precisely controlled in this experiment by manipulating the size and quantity of drug-containing droplets. It could realize modulations of drug doses as low as 200 µg, which was well suited to meet dose control with highly active drugs or drugs with a narrow therapeutic window. A lower-dose drug micro-delivery model could be established by adjusting the model size, printing solution concentration, printing resolution, droplet size, and printing layers.

Combination therapy enhances efficacy and convenience by leveraging the synergy of drugs, thereby increasing patient compliance [73]. Giovanny et al. [74] dispersed lisinopril and spironolactone in light-curing biological inks. They were dispensed through a piezoelectric nozzle onto a drug carrier composed of two attachable compartments fabricated via binder jet 3D printing. Each chamber held 250μL of drug solution, and the subsequent treatment prevented the respective drug solution from being absorbed into the carrier tablet during preparation. Finally, by combining the two solutions in the chamber, a multi-tablet for the treatment of hypertension could be obtained (Figure 11). This study demonstrated the efficacy of binder jet 3D printing in the formulation of combined therapeutic oral dosage forms containing hydrophilic and hydrophobic drugs.

### 3.4. Preparations for Children

For a long time, the pediatric population has been overlooked in developing oral dosage forms. 90% of medications do not come in pediatric dosage forms. The commonly used dosage distribution method for children is mostly to split adult preparations to meet the needs of children. This method’s disadvantages include drug pollution and inaccurate dosage [75]. Wang and colleagues [76] used an isopropanol aqueous solution as a binder to create a color cartoon levetiracetam children’s preparation with an attractive appearance and immediate release characteristics via color inkjet three-dimensional printing (CJ-3DP) (Figure 12). The dose model demonstrated a robust linear relationship between the model volume and tablet strength (>0.999). The results of the surface roughness analysis indicated that the appearance of the CJ-3DP tablet was significantly superior to that of the 3D printing levetiracetam tablet. The example demonstrated how 3D printing technology could create cartoon preparations that improve drug release behavior and attempt to differentiate drug loading using cartoon preparations with various shapes. Additionally, cartoon preparations appeal more to children than standard tablets, which can help alleviate children’s fear of taking medicine and increase medication compliance.

### 3.5. Implants

The binder jet 3D printing process is capable of producing complex geometries with millimeter-scale dimensions. The capability of this technology to create complex geometries has been recently extended to the manufacture of biomaterials in the field of tissue engineering [77]. Implants are a class of agents surgically implanted or introduced into the site of action through a special device. However, traditional implants make it difficult to achieve personalized implantation based on patients’ age, anatomical differences, gender, and disease state. Moreover, they may reduce treatment effectiveness and introduce safety concerns. By contrast, rapid advancements in 3D printing technology foster research and development of implants for personalized treatment. Unlike conventional pressing methods, 3D printing drug-loaded implants can achieve complex release patterns and customized shapes for each patient [78]. Zhou et al. [79] used binder jet 3D printing technology to identify two potential water-soluble adhesives and replaced bone with hydroxyapatite (HA) in the chemical composition phase and natural bone in the inorganic phase (Figure 13). The HA powder was printed into a bone tissue implant to strengthen the bond between the HA powder particles and the bone implant’s HA content. The final experimental results indicated that it exhibited more excellent printability, more geometric accuracy, and more compressive strength with a mixture of high molecular mass polyvinyl alcohol powder and HA than commercial calcium sulfate bone implants. Wang et al. [80] used binder jet 3D printing technology to create polyvinyl alcohol (PVA) and phosphoric acid implants. Compared with the polyvinyl alcohol implants, the phosphoric acid implants had better fabrication accuracy, micro-architecture, and suitable mechanical properties. Their study also demonstrated that different composition ratios of HA and β-Tricalcium phosphate (β-TCP) slightly influenced the implants’ mechanical properties. As the weight ratio of HA to β-TCP in the implants increased, the implants’ compressive strength increased. Considering the implants’ mechanical and biocompatible properties, the phosphoric acid implants with a HA/β-TCP weight ratio of 60:40 may be the best candidate for bone tissue engineering applications. Bose et al. [81] synthesized doped iron and silicon by solid-state synthesis β-TCP powder. They used the binder jet 3D printing technique to prepare porous ferric oxide and silica implants of tricalcium phosphate. The mechanical properties and compressive strength of the implants are outstanding. The presence of iron and silicon promotes bone mineralization and early bone formation in implantation. After 12 weeks, new blood vessels were formed, which promoted wound healing.

In a study by Ji-Ho et al. [82], 3D printing calcium sulfate hemihydrate (CSH) scaffolds were fabricated based on the binder jet 3D printing technique. Then CSH scaffolds were converted to biphasic calcium phosphate (BCP) using hydrothermal and heat treatment. The melted polycaprolactone (PCL) was infiltrated into the resulting BCP scaffold through a capillary rise infiltration process (Figure 14). The PCL/BCP composite scaffold had the highest compressive strength, modulus, and toughness, with a change in fracture mode from brittle to less brittle. In addition, the stable PCL/BCP surface promoted initial cellularity and showed adequate proliferation and differentiation of pre-osteoblastic cells. Bruce L. Tai et al. [83] made a new type of bone-mimicking material, namely a 3D polymer-infiltrated composite. This material consists of plaster powders and epoxy. The plaster part was made by binder jet 3D printing technology and then infiltrated with epoxy to develop the desired strength (Figure 15). Table 2 lists a lot of examples of binder jet 3D printing porous implants for bone tissue regeneration and demonstrates that binder jet 3D printing technology is feasible and has the potential to be developed for the preparation of implants with complex structures.

**Table 2 pharmaceutics-14-02589-t002:** Examples of porous implants for bone tissue regeneration that have been used in recent research on binder jet 3D printing.

Powder	Binder	Layer Thickness (µm)	Reference
α-tricalcium phosphate(α-TCP)	10 wt% phosphoric acid solution	50	A. Butscher [84]
HA/poly(vinyl)alcohol (PVOH)	water-based binder	100	Sophie C. Cox [85]
SiO_2_/Zn-O/β-TCP	Not mentioned	20	Samit Kumar Nandi [86]
Calcium Sulfate hemihydrate	2-pyrrolidinone solution	89	Mitra [87]
hydroxyapatite and a-tricalcium phosphate	phosphate buffer, Tween 80	89	Jason A. [88]
α-TCP	2.5 wt% disodium hydrogen phosphate solution	88	Ruth [89]
HA microsphere	water-based polymeric binder	90	Chai [90]
MgO/ZnO-TCP	Not mentioned	Not mentioned	Dong-xu Ke [91]
Mg–Si-doped TCP	Not mentioned	Not mentioned	SUSMITA BOSE [92]

**Figure 13 pharmaceutics-14-02589-f013:**
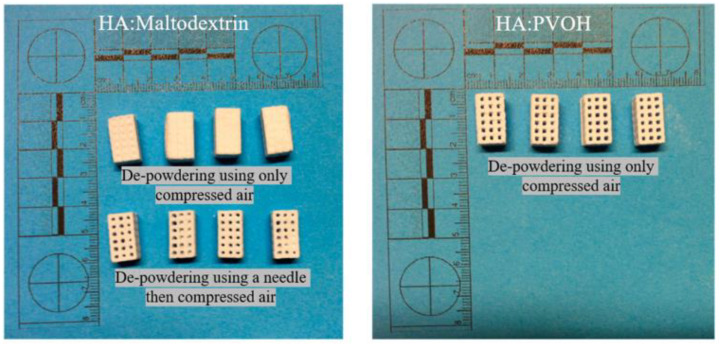
Processing behavior of HA powders mixed with maltodextrin, PVOH (low MW), and PVOH (high MW). After de-powdering, the difference in removing powders from the 3D-printed scaffolds between HA: maltodextrin and HA: PVOH was demonstrated [79].

**Figure 14 pharmaceutics-14-02589-f014:**
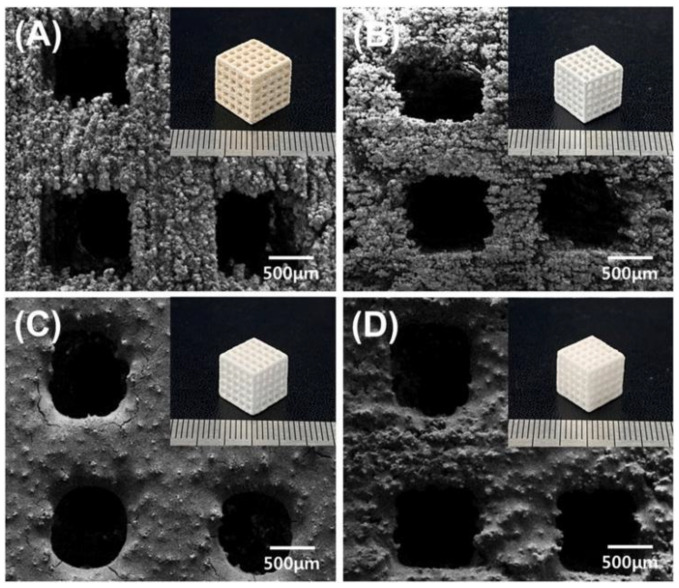
FE-SEM and (inset) optical images of the surface of the 3D-printed (**A**) CSH, (**B**) BCP, (**C**) BCP/_D_-PCL, and (**D**) BCP/_m_-PCL scaffolds [82].

**Figure 15 pharmaceutics-14-02589-f015:**
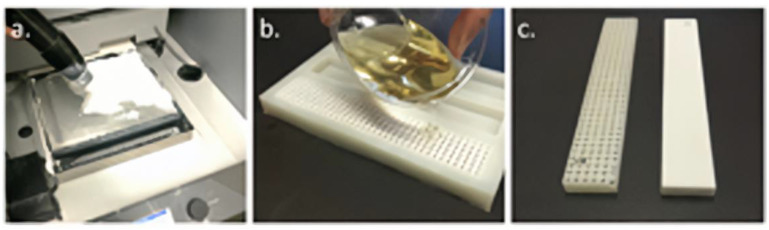
Manufacturing steps for 3DPIC: (**a**) 3D printing, (**b**) epoxy infiltration and casting, and (**c**) cured examples of a 50% 3DPIC (left) and 100% 3DPIC (right) [83].

## 4. Application Prospect and Challenge of Binder Jet 3D Printing Technology

Binder jet 3D printing technology has been maturing over several years of development. Summarizing the above chapters, it can be learned that binder jet 3D printing mainly focuses on fast disintegration and biomedical engineering applications. However, the potential still remains to be explored.

(1)Structural advantages create complex drug delivery systems;

Binder jet 3D printing technology has shown advantages in pharmaceutical product design and manufacturing increasingly complex drug delivery systems, which are difficult to make via conventional technology. Modifying the structure and shape of the model makes it possible to fabricate preparations with personalized structures and customized release mechanisms, which create new opportunities for drug delivery systems. Moreover, with the accuracy and flexibility of binder jet 3D printing, it can be useful in areas such as the administration of drugs to special populations or drugs with high toxicity and narrow therapeutic windows. In addition, binder jetting 3D printing has shown its potential in formulation development [93]. It helps speed up the drug development process and allows testing to be performed faster in the fabrication of drug products.

(2)All-in-one manufacturing and small batches of manufacturing;

Since the implementation of the Precision Medicines Initiative in the United States in 2015, a trend has shifted from “one-size-fits-all” to personalized medicine [94]. The preparation of binder jet 3D printing is distinct from large-scale assembly line production. It enables the development of personalized customization in small batches, optimizes the allocation of medical resources, and lowers production costs and waste of space and time. Additionally, the all-in-one technology reduces the problems associated with complex processes and has a lower-cost drug development process.

### Challenge

While binder jet 3D printing technology has significant potential in developing personalized preparations and realizing accurate and simplified medication, it still faces considerable market barriers. As we all know, regulation is bound to follow when a concept becomes well-known [95]. As binder jet 3D printing technology is still emerging, the printing preparations must meet the current manufacturing and control standards of medical products and equipment.

Safety concerns are paramount for any pharmaceutical formulation and must be addressed. Due to many factors affecting the quality and safety of computer-designed dosage forms, there is an urgent need for appropriate regulations to ensure the safety of patients and operators [96]. In 2014, the FDA established the emerging technology team (ETT) to help approve emerging technologies and products. In 2017, the FDA issued industry guidance to promote emerging technologies in pharmaceutical innovation, listing 3D printing drugs as a strategic direction. As time goes by, the regulatory approvals of 3D printing dosage forms are increasing [97]. Triastek received Investigational New Drug (IND) approval for its first 3D printing drug product, T19, from the FDA in 2021. T19 was printed using Melt Extrusion Deposition (MED) to treat rheumatoid arthritis. T20 as well as comes from Triastek, which received IND approval for its second 3D printing drug product in 2022. There is reason to believe that a perfect supervision and management system will continue to develop in practice. Secondly, in binder jetting 3D printing technology, the binder is usually composed of organic solvents. According to ICH guideline Q3C (R5), solvents have certain acceptable limits. Therefore, the choice of solvents is limited, and each solvent has a minimum allowable residual amount. A residual solvent check is an indispensable step for the sake of patient safety. Thirdly, in the binder jetting 3D printing process, due to its unique printing principle, powders are stacked into tablets, and the lack of a compaction step leads to a rough surface of the preparation and a relative lack of mechanical strength [98]. The printing equipment may also lead to poor mechanical properties of the formulations, such as the problem of the printing nozzles clogging. Therefore, there is an urgent need to improve the mechanical properties of the formulations by optimizing the printing equipment and optimizing the printing process parameters. In addition, personalized preparations are produced in small batches for quality control. If they are inspected in batches, human resources will be wasted. Finally, when the personalized preparation is transferred from the pharmaceutical factory to the hospital pharmacy department, it is necessary to conduct regular training for the personnel involved, which adds to the difficulty of popularizing binder jet 3D printing [99].

## 5. Conclusions

This paper reviews the literature on the binder jet 3D printing process. The application of binder jet 3D printing technology in pharmaceutical manufacturing advances the development of personalized drug delivery, maximizes drug efficacy in patients, and significantly improves patient medication adherence. Although there are still some issues to be resolved, we are confident that by establishing and continuously improving a drug quality surveillance system and advancing technology, we can develop a 3D printing device that meets the requirements for high printing accuracy and gradually resolve current issues. The wave of pharmaceutical manufacturing and reform will be led by 3D printing technology.

## Figures and Tables

**Figure 1 pharmaceutics-14-02589-f001:**
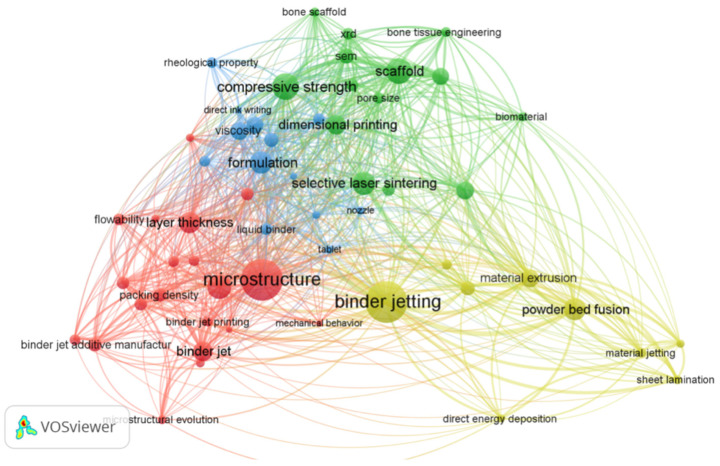
VOS viewer network visualization map of 3D printing/Rapid prototyping/Additive manufacturing technology involved in pharmacy and orthopedics.

**Figure 2 pharmaceutics-14-02589-f002:**
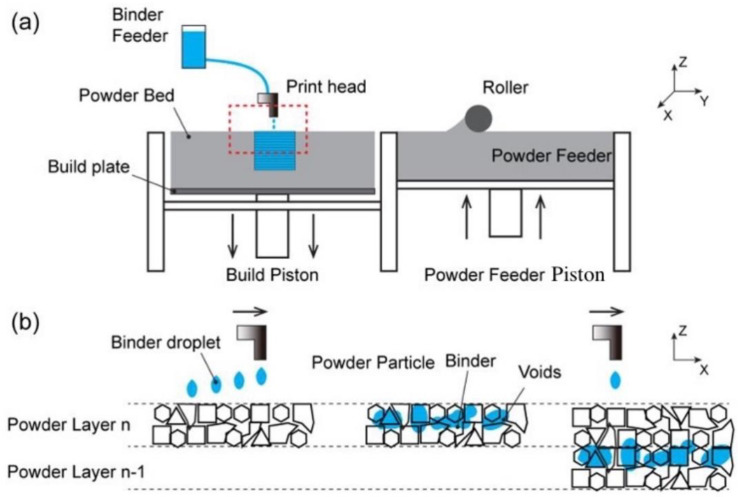
Schematic illustrations of the 3DP process: (**a**) 3DP inkjet printing system, (**b**) Enlargement of the area in red rectangle: powder/binder interaction between adjacent layers [12].

**Figure 3 pharmaceutics-14-02589-f003:**
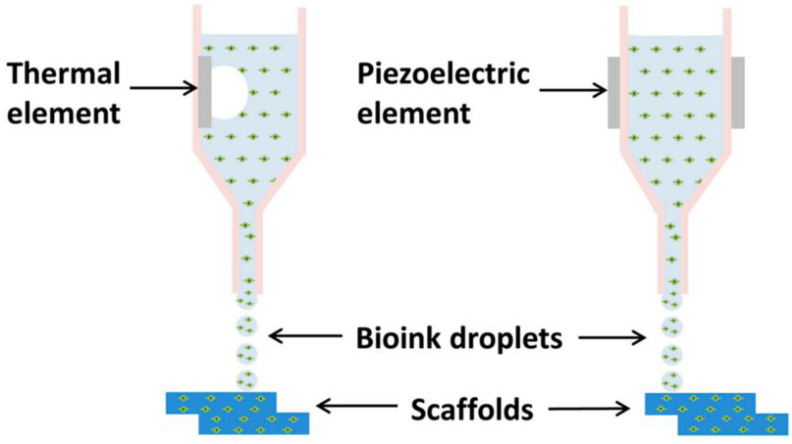
Schematic of Inkjet-based Bioprinting. Thermal inkjet uses heat-induced bubble nucleation that propels the bio-ink through the micro-nozzle. The piezoelectric actuator produces acoustic waves that propel the bio-ink through the micro-nozzle [19].

**Figure 4 pharmaceutics-14-02589-f004:**
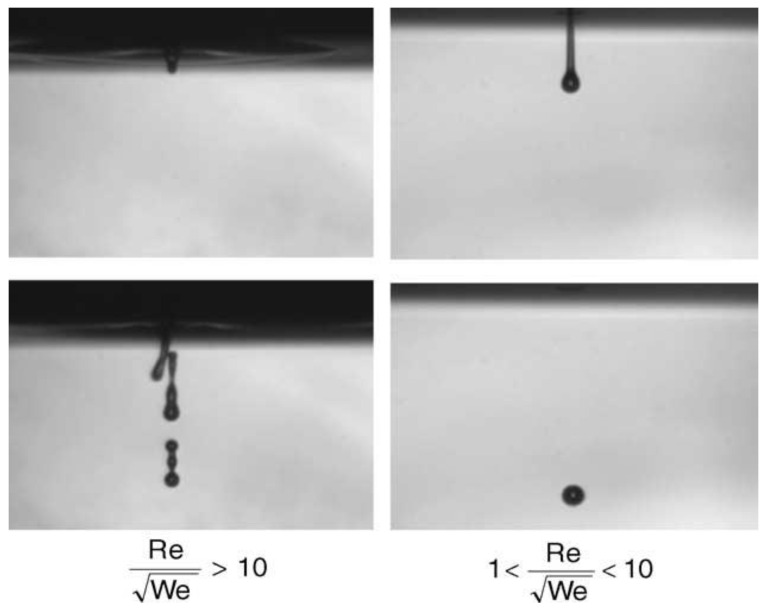
Ejection images of suspensions showing the effect of the ratio Re/We^1/2^ (Z value) [25].

**Figure 5 pharmaceutics-14-02589-f005:**
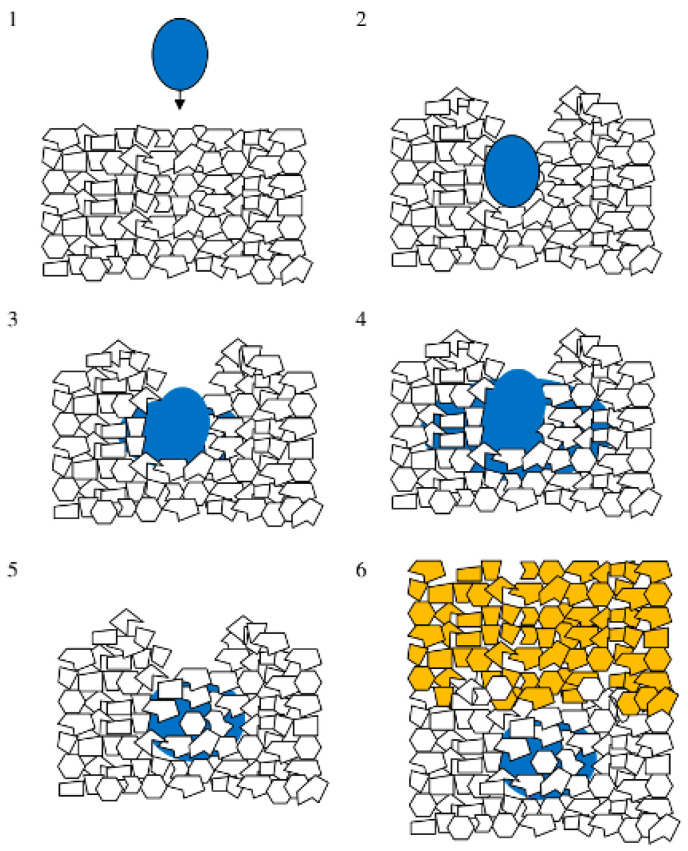
Schematic illustration of the main stages of single binder drop powder interaction during 3DP: build-up of a homogeneous thin powder layer (step 1); binder droplet delivery on the powder bed while maintaining its integrity (step 2); wetting of the powder by the binder (step 3); spreading of the drop within the powder (step 4); binder/powder reaction and hardening (step 5); recoating with a new powder layer (step 6); extraction of the green specimen from the powder bed (step 7, not depicted); removal of loose powder within the green specimen (step 8, not pictured) [35].

**Figure 6 pharmaceutics-14-02589-f006:**
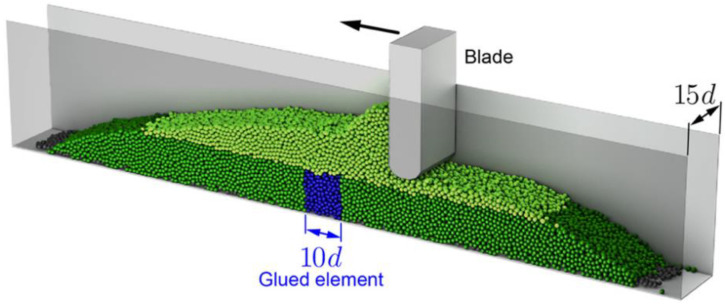
Discrete Element model of powder bed and blade powder spreading [45].

**Figure 7 pharmaceutics-14-02589-f007:**
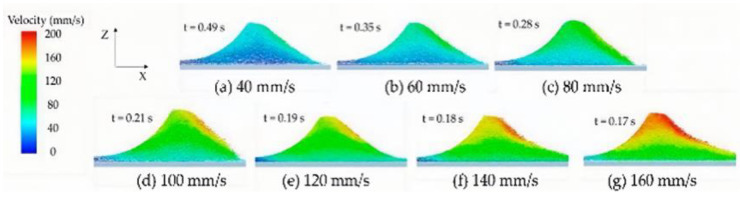
Powder flow velocity under different roller’s translational velocity [46].

**Figure 8 pharmaceutics-14-02589-f008:**
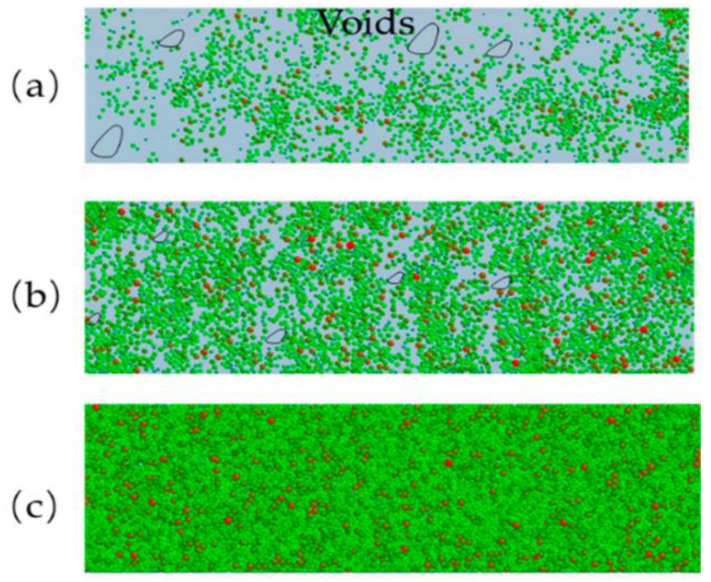
The powder-bed characteristics under different layer thickness values: (**a**) H = 50 μm; (**b**) H = 75 μm; and (**c**) H = 150 μm [46].

**Figure 9 pharmaceutics-14-02589-f009:**
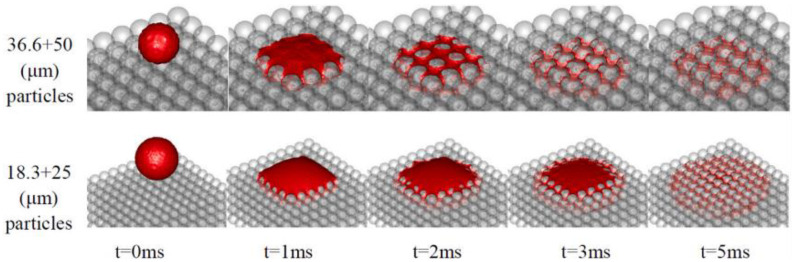
Droplet impact and penetration into the two types of powder beds [47].

**Figure 10 pharmaceutics-14-02589-f010:**
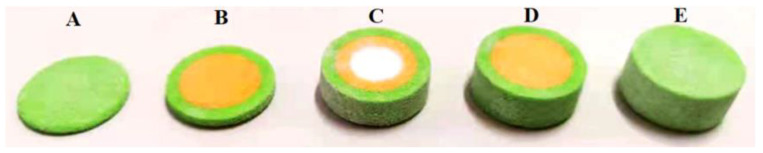
Structure diagram for the 3D-printed compound LEV-PN multi-compartmental structured dispersible tablet. (**A**): 1–7 layers; (**B**): 1–13 layers; (**C**): 1–38 layers; (**D**): 1–44 layers; (**E**): 1–50 layers [72].

**Figure 11 pharmaceutics-14-02589-f011:**
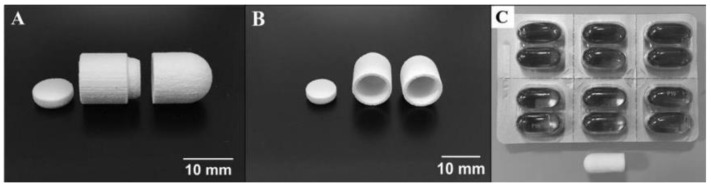
Multi-compartment preform tablet. (**A**) Side view of the three pieces constituting the preform tablet. From left to right, top cap, top compartment, and bottom compartment. These pieces were manufactured by powder-bed 3D printing using binder DI water with 5% ethanol and 0.25% Tween 80. The powder utilized for their construction was calcium sulfate. Each of the two wells has a 250 µL capacity. (**B**) Front view of the preform tablet fabricated. (**C**) Comparison between the assembled version of the preform tablet and commercially available gel capsules [74].

**Figure 12 pharmaceutics-14-02589-f012:**
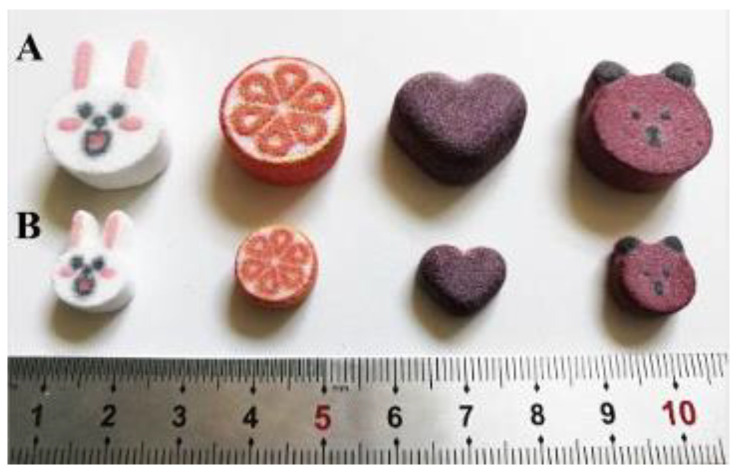
3D printing of colorful cartoon dispersion tablets (**A**): 1000 mg strength; (**B**): 250 mg strength [76].

**Table 1 pharmaceutics-14-02589-t001:** Examples of rapid-release formulations that have been used in recent research on binder jet 3D printing.

Dosage form	API	Powder	Binder	Reference
Instant-Dissolving tablets	Levetiracetam	Microcrystalline cellulose, mannitol, Colloidal silicon dioxide, polyvinyl pyrrolidone	40% (*v/v*) isopropanol aqueous solution containing 0.05% (*w*/*w*) PVP and 4% (*w/w*) glycerin	Wang [58]
Dispersible tablets	ketoprofen	lactose monohydrate, spray-dried lactose monohydrate, microcrystalline cellulose, mannitol, polyvinyl pyrrolidone grade K 25, silica	Ethanol solution with 10% polyethylene glycol 1500	Klemen Kreft [59]
Fast-Dissolving tablets	Indomethacin	Lactose monohydrate, Kollidon^®^VA64 (KL)	5% (*w/v*) KL in water	Chang [60]
Oral disintegrating tablets	Warfarin sodium	D-sucrose, pregelatinized starch, povidone K30, Microcrystalline cellulose, silicon dioxide	38% (*v*/*v*) Ethanol solution	Tian [61]
Dispersible tablets	Clozapine	Mannitol, lactose, microcrystalline cellulose, strawberry flavor, Colloidal silicon dioxide	50% (*v*/*v*) Ethanol solution containing 0.3% (*w*/*w*) PVP and 4% (*w/w*) glycerin	CHEN [62]
Orally disintegrating tablets	Andrographolide	Sucrose, mannitol, PVP K30, microcrystalline cellulose, aspartame	30% (*v*/*v*) Ethanol solution	HUANG [63]

## Data Availability

Not applicable.

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
