# Peer review of "The Application and Challenge of Binder Jet 3D Printing Technology in Pharmaceutical Manufacturing"

_pharmaceutics, 2022, doi:10.3390/pharmaceutics14122589_

Round 1

Reviewer 1 Report

The authors made a review of the current literature of Binder Jet 3DP which is a hot topic of pharmaceutical research. The paper summarizes the state of the art of the field correctly, but I barely miss a real comprehensive evaluation of the cited research. The extension of Chapter 4.1. may be a good way for the evaluation, but in current form it contains no new information compared with the previous chapters, and may be deleted from the text.

The re-use of previously published Figures may be also problematic have the authors permission for the use of pictures?

The paper also contain a lot of mistyping please check the whole paper carefully. Below is a long but not full list of the observed problems:

Minor comments:

Figure 2 powder feeder piston instead of pistion

line 106 the instead of The

line 288 a space is missing before (Figure 6)

line 332, I disagree with the sentence that directly compressed or wet granulated systems cannot meet simultaneously with criteria of ease of administration, rapid drug absorption and high-bioavailability

Lines 357 and 362 spaces are missing before or acter the reference citation.

Line 368 a hollow row would be good to separate Table legend from the normal text

Table 2 instead of TABLE 2

line 535 Binder instead of binder

Reviewer 2 Report

This review covers important aspects of binder jetting 3D printing. Although the authors include a lot of relevant literature in this review, I think that overall quality is not sufficient for publication in the present form.

The main flaw of this paper is poor language quality. There are some mistakes and some sentences are difficult to understand. Also, the authors used some terms not common in the relevant scientific literature.

Line 89-90: The two-dimensional image or image is converted to three-dimensional printing data using computer-aided design (CAD).

This is rather oversimplified interpretation of 3D printing. 3D object design is created by CAD software which is further converted to .stl or some other format compatible with printers. Before printing .stl file is subjected to slicing which converts object to 2D layers and create instruction for printing in G-code.

Line 120: When the heat-sensitive drugs or excipients, it will affect the properties of the binder.

This rather affects stability of heat labile compounds. However, considering the duration of heat exposure, overall risk is low in most cases.

Line 214: Fluidity of the powder should be replaced with flowability.

Line 233-235. The authors neglected the fact that decreasing of the porosity will negatively affect binder penetration and in some cases may result in inadequate mechanical characteristics of the product.

The more muscular the mechanical strength of the preparation – this is very uncommon terminology for mechanical characteristics of the tablets.

Line 244 the molding effect of the preparation. This term is more adequate for some other 3D printing technique and not for binder jetting.

Line 309: smaller particle bed. Did you mean smaller particle size or thinner powder bed?

Line 332. Powder compression and wet granulation are more adequate that powder tablet pressing and wet granulation tablet pressing, respectively.

Line 339. When printing tablets with varying doses of the target drug, the accuracy ranges from 91.2 to 108 percent, and the coefficient of variation ranges from 0.5 to 3.2 percent.

Did you mean on the drug content, or something else?

Dispersible tablets should be used instead of dispersive

Line 380. Easy residue as the binder?????????

Very unclear terminology.

Line 382-384. Yu DG 382 et al. [67] developed a pie-shaped controlled-release tablet for insoluble drugs. Acetaminophen was used as the model drug.

There is some mistake, since acetaminophen is highly water-soluble.

Line 394. the tablet's dissolution volume can also remain relatively stable

Very unclear.

What did you mean under compound preparation? I think that fabrication of dosage forms with multiple drugs is more adequate term.

Line 410-412. These two sentences should be merged.

Line 436. blank preform tablet. Very unclear and uncommon.

There is no reason to give separate section to Chinese drugs, since their characteristics that limit fabrication of dosage forms are also characteristic to some conventional APIs. There is nothing more complex in Chinese drugs.

Round 2

Reviewer 1 Report

The authors worked a lot on the correction of mistypings and highly improved the readability and scientific soundness of the paper, but some of my previous concerns, such as that chapter 4.1 contains some kind of summary of the previous chapters, and no real evaluation of the technological gaps of the previously published research is given. This would give some novelty to the content of the paper and would help the identification of the deficiencies of the current knowledge and designate of future research directions.

Some mistipngs such as the small letter start of the title of chepters 3.3. and 3.4 remained in the text.

Reviewer 2 Report

The authors seriously considered all my comments and made significant efforts to improve the quality of the manuscript. In my opinion, the paper can be accepted. There is only one minor suggestion:

Line 337 The accuracy of printing tablets with target drug content was 91.2 ~ 108%,  with a slight variation coefficient in the 0.5 ~ 3.2%.

Please change to The dose accuracy of the printed tablets

Round 3

Reviewer 1 Report

The authors mostly responded all of my previous concerns, neverteless some minor correction, such as double check of the reference surnames in Table 2 should be still corrected prior to publication.